# Intakes of Folate, Vitamin B6, and Vitamin B12 in Relation to All-Cause and Cause-Specific Mortality: A National Population-Based Cohort

**DOI:** 10.3390/nu14112253

**Published:** 2022-05-27

**Authors:** Yacong Bo, Huadong Xu, Huanhuan Zhang, Junxi Zhang, Zhongxiao Wan, Xin Zhao, Zengli Yu

**Affiliations:** 1The Fifth Affiliated Hospital, School of Public Health, Zhengzhou University, Zhengzhou 450000, China; boyacong@163.com (Y.B.); zhanghh@zzu.edu.cn (H.Z.); zhxwan@suda.edu.cn (Z.W.); 2School of Public Health, Hangzhou Medical College, Hangzhou 310013, China; xuhuadong@hmc.edu.cn; 3NHC Key Laboratory of Birth Defects Prevention, Henan Key Laboratory of Population Defects Prevention, Zhengzhou 450000, China; zhangjunxi0378@126.com; 4The Third Affiliated Hospital of Zhengzhou University, Zhengzhou 450000, China

**Keywords:** diet, folate, vitamin B6, vitamin B12, mortality

## Abstract

The evidence regarding the intake of dietary folate, vitamin B6, and vitamin B12 in relation to mortality in the general population is limited. This study aimed to examine the relationship between dietary intakes of folate, vitamin B6, and vitamin B12 in relation to all-cause and cause-specific mortality in a large U.S. cohort. This study included a total of 55,569 adults from the Third National Health and Nutrition Examination Survey (NHANES III) and NHANES 1999–2014. Vital data were determined by linking with the National Death Index records through 31 December 2015. Cox proportional hazards models were used to investigate the relationships of all-cause and cause-specific mortality with dietary folate, vitamin B6, and vitamin B12 intake. Dietary intakes of folate and vitamin B6 were inversely associated with mortality from all-cause, cardiovascular disease, and cancer for men and with mortality from all-cause and cardiovascular disease for women. In men, the multivariable hazard ratios (95% confidence intervals) for the highest versus lowest quintiles of folate and vitamin B6 were 0.77 (0.71–0.85) and 0.79 (0.71–0.86) for all-cause mortality, 0.59 (0.48–0.72) and 0.69 (0.56–0.85) for CVD mortality, and 0.68 (0.56–0.84) and 0.73 (0.60–0.90) for cancer mortality, respectively. Among women, the multivariable hazard ratios (95% confidence intervals) for the highest versus lowest quintiles of folate and vitamin B6 were 0.86 (0.78–0.95) and 0.88 (0.80–0.97) for all-cause mortality and 0.53 (0.41–0.69) and 0.56 (0.44–0.73) for CVD mortality, respectively. No significant associations between dietary vitamin B12 and all-cause and cause-specific mortality were observed. In conclusion, higher dietary intakes of folate and vitamin B6 were significantly associated with lower all-cause and cardiovascular mortality. Our findings suggest that increasing the intake of folate and vitamin B6 may lower the mortality risk among U.S. adults.

## 1. Introduction

Folate, vitamin B6, and vitamin B12 are essential B vitamin nutrients which play important roles in the degradation of homocysteine (Hcy) by acting as prerequisite substrate donors or as necessary coenzymes [1,2]. In addition, these B vitamins are important components of one-carbon metabolism, which contributes to DNA synthesis and DNA methylation, the synthesis of blood cells, and nerve function. The lack of these B vitamins is one of the key contributors to high blood homocysteine levels [3], DNA repair disruption [4], and gene expression [5].

Several epidemiologic studies have investigated the relationship between dietary folate, vitamin B6, or vitamin B12 intake and mortality risk. However, most of these studies were conducted among specific populations (e.g., older adults and participants with cancers), and limited information is available for general populations. In addition, the results were inconsistent. For example, Persson et al. [6] reported inverse associations between dietary folate and the risk of all-cause and liver-disease mortality in the American Association of Retired Persons Diet and Health Study. Li et al. observed an inverse association between dietary folate intake and all-cause mortality among patients with breast cancer [7]. Medrano et al. observed that a higher dietary intake of folate or vitamin B12 was associated with lower coronary mortality, but vitamin B6 intake was not associated with cardiovascular-disease (CVD) mortality in the Spanish population [8]. Cui et al. [9] observed in the Japan Collaborative Cohort Study that dietary folate and vitamin B6 intakes were inversely associated with mortality from stroke, coronary heart disease, and heart failure, while dietary vitamin B12 intake was positively associated with mortality from stroke in men. In contrast, Xu et al. [10] reported that dietary folate, vitamin B6, and vitamin B12 were neither associated with all-cause mortality nor cancer–specific mortality in participants with cancer. Non-significant relationships between folate supplementation and all-cause or CVD mortality were observed in a recent meta-analysis of randomized clinical trials (RCTs) [11], and even some RCTs demonstrated that folate or vitamin B12 supplementation was associated with higher all-cause mortality [12,13,14]. Large-scale cohort studies are thus needed to provide robust results and precise estimates in general populations. Therefore, utilizing the data from a nationally representative sample of the National Health and Nutrition Examination Survey (NHANES), we investigated whether dietary intakes of folate, vitamin B6, and vitamin B12 were in relation to the risk of all-cause and cause-specific mortality in the general U.S. adult population.

## 2. Materials and Methods

### 2.1. Study Population

The NHANES is a nationally representative survey conducted by the National Center for Health Statistics of the Centers for Disease Control and Prevention (CDC). Information on the methods of sampling and data collection has been described previously [15]. In brief, the NHANES adopted a complex, stratified, multistage probability sampling method to select a representative population in the U.S. Information on lifestyle factors and health and nutritional status was collected for each participant. The design and procedures of the NHANES survey have been described in detail elsewhere [16].

In the current study, we used data from NHANES III (1988–1994) and eight cycles of the NHANES from 1999 and 2014. Adult participants aged ≥ 20 years were included in the analysis. Figure 1 shows the procedure of participant selection. Briefly, 113,402 U.S residents attending the medical examination in the NHANES from 1988 to 2014 were initially included in this study. Among them, 52,579 participants were excluded due to them being younger than 20 years old, and 5254 participants with missing information on dietary folate, vitamin B6, or vitamin B12 intake were further excluded. Finally, 55,569 adults were included in the main analysis.

### 2.2. Measurement of Dietary Folate, Vitamin B6, and Vitamin B12 Intake

The data on dietary folate, vitamin B6, and vitamin B12 intake from food were assessed using a 24 h diet recall conducted by trained interviewers. In NHANES III and the 1999–2002 survey cycles, one diet recall was collected from each person in the Mobile Examination Center (MEC). Since 2003, two dietary reviews were conducted. The first one was collected in the MEC, and the second collection was conducted by telephone after 3–10 days. The intakes of folate, vitamin B6, and vitamin B12 from food were evaluated using the United States Department of Agriculture (USDA) Food and Nutrient Database for Dietary Studies (FNDDS), versions 1.0–5.0, which provided the content of these vitamins in each food [17].

### 2.3. Ascertainment of Mortality

The NCHS has linked the NHANES survey data with all-cause and cause-specific mortality from the National Death Index (NDI). The restricted-use Linked Mortality File (LMF) has been updated with mortality follow-up data through 31 December 2015, which was also publicly available. The disease-specific death was determined by the 10th revision of the International Classification of Disease (ICD-10). The outcomes in this study were all-cause, cardiovascular, and cancer mortality. Cardiovascular mortality was defined as the ICD-10 codes of I00–I09, I11, I13, I20–I51, or I60–I69, and cancer mortality was defined as the ICD-10 codes of C00–C97. The time-to-event for each individual was calculated from the date of recruitment to the date of death or the censor date (31 December 2015), whichever came earlier.

### 2.4. Covariates

We used the information on age, sex (men or women), race/ethnicity (Hispanic, non-Hispanic white, non-Hispanic black, and other race—including multi-racial), smoking (never, former, or current), alcohol drinking (nondrinker, low-to-moderate drinker, or heavy drinker), the ratio of family income to poverty (≤1, 1–3, or >3), leisure time physical activity (defined as the product of metabolic equivalent value (MET)), and self-reported physician diagnosed cardiovascular disease (yes or no), hypertension (yes or no), type 2 diabetes (yes or no), and Body mass index (BMI) as the covariates.

### 2.5. Statistical Analysis

We used numbers (percentages) to depict the categorical variables and means (standard deviations) to describe the continuous variables. Multiple imputation was used to impute these covariates with missing values [18].

The Cox proportional hazards model was adopted to evaluate the relationships of dietary folate, vitamin B6, and vitamin B12 with all-cause and cause-specific mortality. The dietary nutrients (i.e., dietary folate, vitamin B6, and vitamin B12) were categorized into quartiles, and the first quartile was selected as the reference group. Two models were introduced: the Crude Model, which was not adjusted, and the Multivariable Model, which was adjusted for age, race/ethnicity, BMI, family income–poverty ratio, smoking status, drinking status, leisure-time physical activity, total energy intake, diabetes, hypertension, and cardiovascular disease. 

Stratified analyses were conducted to evaluate the potential effect modification by age (<65 or ≥65 y), smoking status (yes or no), drinking status (yes or no), and BMI (<30 or ≥30 kg/m^2^). We examined each potential modifier separately by adding a multiplicative interaction term (i.e., continuous dietary nutrients intake parameters* potential modifier).

In order to examine the robustness of the relationships, we also conducted three sensitivity analyses: (1) restricting the analysis to participants free of cardiovascular disease at baseline; (2) excluding participants who died within the first 2 years of follow-up; (3) further adjusting the folate supplement.

All statistical analyses were conducted using R 4.0.2. (R Core Team, Vienna, Austria), and two-sided *p* values < 0.05 were considered statistically significant.

## 3. Results

### 3.1. General Characteristics

The general characteristics of the participants and the cases of all-cause death are presented in Table 1. A total of 55,569 participants with a mean age of 49.0 (SD: 18.6) years at baseline were included. During a median follow-up of 9.8 (interquartile range (IQR): 5.3–15.8) years, a total of 11,535 participants died, including 2411 deaths due to cardiovascular disease and 2482 deaths due to cancer. In comparison to the survivors, the dead participants were more likely to be older, female, and non-Hispanic white; more likely to have diabetes, hypertension, and cardiovascular disease; less likely to be current smokers; and had a lower family income and less leisure-time physical activity.

### 3.2. Dietary Folate with All-Cause and Cause-Specific Mortality

The sex-specific relationships between dietary folate intake and all-cause and cause-specific mortality are presented in Table 2. Among men, we observed that dietary folate intake was inversely associated with death from all-cause, cardiovascular disease, and cancer. The multivariable hazard ratios (HRs) with 95% confidence intervals (CIs) for the highest versus lowest quintiles in men were 0.77 (0.71–0.85), 0.59 (0.48–0.72), and 0.68 (0.56–0.84) for all-cause, cardiovascular, and cancer mortality, respectively. Consistently, among women, dietary folate intake was inversely associated with all-cause and cardiovascular mortality. However, no significant relationship between dietary folate intake and cancer mortality was observed. The multivariable HRs (95% CIs) for the highest versus lowest quintiles in women were 0.86 (0.78–0.95), 0.53 (0.41–0.69), and 0.82 (0.66–1.03) for all-cause, cardiovascular, and cancer mortality, respectively.

### 3.3. Dietary Vitamin B6 with All-Cause and Cause-Specific Mortality

The sex-specific relationships between dietary vitamin B6 intake and all-cause and cause-specific mortality are presented in Table 3. Among men, a higher dietary intake of vitamin B6 was associated with a lower risk of all-cause (HR (95% CI) 0.79 (0.71–0.86)), CVD (HR (95% CI) 0.69 (0.56–0.85)), and cancer mortality (HR (95% CI) 0.73 (0.60–0.90)). Among women, a higher dietary intake of vitamin B6 was associated with a lower risk of all-cause (HR (95% CI) 0.88 (0.80–0.97)) and CVD mortality (HR (95% CI) 0.56 (0.44–0.73)). No significant relationship between dietary vitamin B6 intake and cancer mortality was observed.

### 3.4. Dietary Vitamin B12 with All-Cause and Cause-Specific Mortality

Table 4 shows the HRs for dietary vitamin B12 intake in relation to all-cause and cause-specific mortality. We observed that vitamin B12 intake was not associated with death from all-cause, cardiovascular, or cancer mortality in either men or women. For men, the multivariable HRs (95% CIs) for the highest versus lowest quintiles were 1.01 (0.92–1.11), 0.93 (0.76–1.13), and 0.96 (0.79–1.16) for all-cause, cardiovascular, and cancer mortality, respectively. For women, the multivariable HRs (95% CIs) for the highest versus lowest quintiles were 1.07 (0.97–1.18), 1.05 (0.83–1.34), and 1.05 (0.84–1.31) for all-cause, cardiovascular, and cancer mortality, respectively.

### 3.5. Subgroup and Sensitivity Analysis

The results of the subgroup analyses are presented in Appendix A. The relationships of dietary folate, vitamin B6, and vitamin B12 with all-cause and cause-specific mortality appeared to be similar to the main results, although several subgroup findings did not reach statistical significance. Statistically significant interactions were observed for some factors. For male participants, we observed stronger negative associations of folate and vitamin B6 intake with cardiovascular mortality in older adults than in younger adults. For female participants, a statistically significant interaction was observed for the relationship between dietary vitamin B12 and cardiovascular mortality in the analyses stratified according to BMI at baseline (*P* for interaction = 0.008), even though the difference did not reach statistical significance in both groups.

Sensitivity analyses suggested that the significant associations of dietary folate and vitamin B6 with all-cause and cause-specific mortality remained similar after excluding the participants who died within the first 2 years of follow-up (Appendix A) and the participants with cardiovascular disease at baseline (Appendix A).

## 4. Discussion

To the best of our knowledge, this is the largest study so far to comprehensively investigate the relationships of the dietary intakes of folate, vitamin B6, and vitamin B12 with all-cause and cause-specific mortality in a large, nationally representative U.S. adults cohort. We found that dietary folate and vitamin B6 intakes were inversely associated with mortality from all-cause, cardiovascular disease, and cancer for men and with mortality from all-cause and cardiovascular disease for women. No significant associations were seen between the intake of vitamin B12 and all-cause and cause-specific mortality in either men or women.

Our findings of the inverse association between dietary folate intake and all-cause and cause-specific mortality are in line with several previous studies. The Swedish Mammography Cohort suggested that the dietary intake of folate was associated with a reduced risk of all-cause mortality in women diagnosed with breast cancer [19]. The Japan Collaborative Cohort Study found that the dietary intake of folate was associated with a reduced risk of mortality from heart failure for men and of mortality from stroke, CHD, and total cardiovascular disease for women [9]. One study from the same cohort suggested that folate intake was associated with lower all-cause and CVD mortality among participants at a high risk of CVD [20]. One ecological study in Spain found that folate intake was associated with a reduced risk of death from ischemic heart disease and cerebrovascular disease in men and of death from cerebrovascular disease in women [8]. However, some other studies have inconsistent results. The American Association of Retired Persons Diet and Health Study did not find a significant association between folate intake and liver disease mortality [6]. Another study in the U.S. reported that dietary folate was neither associated with all-cause nor breast-cancer-specific mortality [10]. This inconsistency can be ascribed to a series of factors, including the variations in the health outcomes, research methods, study populations, and study regions. 

Previous data on the associations between dietary vitamin B6 intake and mortality risk have been limited. Our findings of inverse relationships of dietary vitamin B6 intake with all-cause, cardiovascular, and cancer mortality for men and with all-cause and cardiovascular mortality for women are in line with some previous studies. Three prospective cohort studies found that a higher intake of dietary vitamin B6 was associated with a lower risk of all-cause mortality [21], cardiovascular mortality [9], and prostate cancer mortality [22]. Contrary to our findings, an ecological analysis in Spain did not observe a significant association between dietary vitamin B6 intake and cardiovascular mortality [8]. Among 1508 women with breast cancer, Xu et al. [10] reported non-significant associations between dietary vitamin B6 intake and all-cause or breast-cancer-specific mortality. In a Chinese study, dietary vitamin B6 intake was inversely associated with all-cause and cardiovascular mortality, but it was not associated with cancer mortality [23]. A recent study in U.K. reported that dietary vitamin B6 intake was not statistically associated with colorectal-cancer-specific mortality [24]. Of these previous studies, one was an ecological analysis, three were among participants with cancer, and the other three were among middle-aged or older adults. Thus, it may be difficult to compare them directly with our study.

To date, only seven studies have investigated the relationships between dietary vitamin B12 intake and the risk of death. Four studies reported that a higher intake of dietary vitamin B12 was associated with a lower risk of death from cerebrovascular disease in the general population [8], as well as from all-cause [25] and cancer [25,26] among participants with cancer, which is similar to our findings. However, the three other studies found no significant associations for all-cause [10], breast-cancer-specific [10], prostate-cancer-specific [22], or colorectal-cancer-specific mortality [24]. More interestingly, the Japan Collaborative Cohort Study observed a positive association between dietary vitamin B12 intake and stroke mortality in older men [9]. Again, a direct comparison of these studies is difficult considering the heterogeneity of the study populations and study methods. 

Our study has a series of strengths. Firstly, this study was based on a reliable and robust design with a complex, stratified, multistage probability sampling method, which made it possible to select a representative sample of the general U.S. population. Secondly, the relatively large sample size enabled us to provide more stable and precise estimates and made it possible for us to conduct a series of subgroup and sensitivity analyses. Thirdly, given the comprehensive data collection in the NHANES, we could adjust for a variety of potential confounders, including race/ethnicity, socioeconomic status, lifestyle factors, and comorbidities. 

It should also be acknowledged that out study has several limitations. Firstly, the dietary nutrients intake was collected via one or two 24 h diet recalls for each person, which may not reflect the long-term intake since food intake varies from day to day. However, the National Cancer Institute method was used to reduce the measurement bias due to dietary intake estimation using 24 h diet recalls [27,28]. In addition, this measurement bias seems to be random, and there is no evidence suggesting that the potential measurement bias is different between the participants who survived and those who died. We thus speculate that this limitation should not affect the study’s conclusion. Secondly, the mortality outcomes were determined by linkage to the National Death Index through a probabilistic match, which might result in misclassification. However, a prior validation study showed that 99.4% of the living participants and 96.1% of the decedents were classified correctly [29]. Thirdly, we could not investigate the effect for cancer-cause-specific mortality due to the limited information. Fourthly, we used the pre-existing data analyses of the relationships of diet folate, vitamin B6, and vitamin B12 intake with all-cause and cause-specific mortality, which may not be considered initially in the NHANES survey. In addition, the biomarkers of vitamins were not considered. However, previous studies suggested that diet intake was highly correlated with B vitamin status [30,31,32].

## 5. Conclusions

In summary, using data from a nationally representative survey of U.S. adults, we found that dietary folate and vitamin B6 intakes were inversely associated with mortality from all-cause, cardiovascular disease, and cancer for men and with mortality from all-cause and cardiovascular disease for women. Our findings suggest that increasing the intake of folate and vitamin B6 may lower the mortality risk among U.S. adults.

## Figures and Tables

**Figure 1 nutrients-14-02253-f001:**
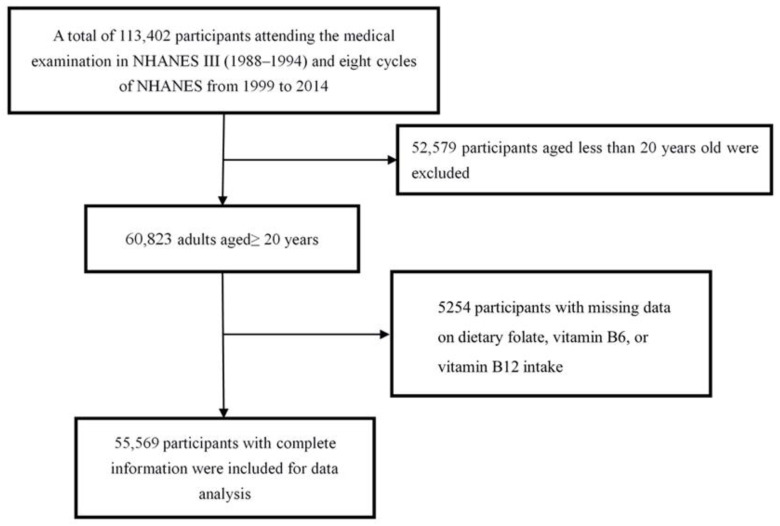
Flowchart of participant selection.

**Table 1 nutrients-14-02253-t001:** Baseline characteristics of the study population.

Variables	Total Sample *n* = 55,569	Survivors *n* = 44,034	Death *n* = 11,535
**Age, year**	49.0 (18.6)	44.4 (16.5)	66.4 (15.7)
**Sex**			
Female	26,566 (47.81%)	20,343 (46.2%)	6223 (53.95%)
Male	29,003 (52.19%)	23,691 (53.8%)	5312 (46.05%)
**Race**			
Hispanic	14,402 (25.92%)	12,011 (27.28%)	2391 (20.73%)
Non-Hispanic white	25,249 (45.44%)	18,973 (43.09%)	6276 (54.41%)
Non-Hispanic black	12,658 (22.78%)	10,090 (22.91%)	2568 (22.26%)
Other race—including multi-racial	3260 (5.87%)	2960 (6.72%)	300 (2.6%)
**Drinking**			
Never	14,732 (26.51%)	10,410 (23.64%)	4322 (37.47%)
Low to moderate	11,607 (20.89%)	8984 (20.4%)	2623 (22.74%)
Heavy	29,230 (52.60%)	24,640 (55.96%)	4590 (39.79%)
**Smoking**			
Never	12,662 (22.79%)	10,050 (22.82%)	2612 (22.64%)
Former	13,919 (25.05%)	9779 (22.21%)	4140 (35.89%)
Current	28,988 (52.17%)	24,205 (54.97%)	4783 (41.47%)
**Ratio of family income to poverty**			
≤1	12,048 (21.68%)	9353 (21.24%)	2695 (23.36%)
1–3	24,021 (43.23%)	18,178 (41.28%)	5843 (50.65%)
>3	19,500 (35.09%)	16,503 (37.48%)	2997 (25.98%)
**Diabetes**	6469 (11.64%)	3975 (9.03%)	2494 (21.62%)
**Hypertension**	10,078 (18.14%)	5814 (13.2%)	4264 (36.97%)
**Cardiovascular disease**	5705 (10.27%)	2806 (6.37%)	2899 (25.13%)
**BMI, kg/m^2^**	28.1 (6.0)	28.2 (6.0)	27.5 (5.7)
**Leisure-time physical activity, MET**	18.5 (28.3)	19.2 (29.0)	15.5 (25.5)

Data were presented as means (standard deviations) or numbers (percentages). BMI, body mass index; MET, metabolic equivalent value.

**Table 2 nutrients-14-02253-t002:** Hazard Ratios (HRs) and 95% confidence intervals (CIs) for all-cause and cause-specific mortality according to the quintiles of dietary folate.

	Quartile of Nutrient Intake				*P_trend_*
	Q1	Q2	Q3	Q4	
**Men**					
**All-cause mortality**					
Crude Model	Ref	0.89 (0.84–0.95)	0.77 (0.72–0.83)	0.65 (0.60–0.71)	<0.001
Multivariable Model	Ref	0.91 (0.86–0.97)	0.86 (0.80–0.92)	0.77 (0.71–0.85)	<0.001
**Cardiovascular mortality**					
Crude Model	Ref	0.80 (0.71–0.91)	0.64 (0.55–0.75)	0.53 (0.44–0.63)	<0.001
Multivariable Model	Ref	0.81 (0.71–0.92)	0.71 (0.61–0.83)	0.59 (0.48–0.72)	<0.001
**Cancer mortality**					
Crude Model	Ref	0.82 (0.72–0.93)	0.77 (0.66–0.89)	0.57 (0.48–0.68)	<0.001
Multivariable Model	Ref	0.87 (0.77–0.99)	0.89 (0.77–1.04)	0.68 (0.56–0.84)	0.001
**Women**					
**All-cause mortality**					
Crude Model	Ref	1.01 (0.94–1.07)	0.91 (0.84–0.98)	0.85 (0.78–0.93)	<0.001
Multivariable Model	Ref	0.91 (0.85–0.97)	0.87 (0.81–0.95)	0.86 (0.78–0.95)	<0.001
**Cardiovascular mortality**					
Crude Model	Ref	0.91 (0.79–1.05)	0.71 (0.59–0.86)	0.58 (0.46–0.71)	<0.001
Multivariable Model	Ref	0.82 (0.70–0.95)	0.69 (0.57–0.83)	0.53 (0.41–0.69)	<0.001
**Cancer mortality**					
Crude Model	Ref	0.97 (0.84–1.13)	1.04 (0.88–1.24)	0.79 (0.65–0.97)	0.115
Multivariable Model	Ref	0.93 (0.80–1.08)	1.05 (0.88–1.25)	0.82 (0.66–1.03)	0.342

Crude HR did not adjust for anything. Multivariable HR adjusted for age, race/ethnicity, BMI, family income–poverty ratio, smoking status, drinking status, leisure-time physical activity, total energy intake, diabetes, hypertension, and cardiovascular disease.

**Table 3 nutrients-14-02253-t003:** Hazard Ratios (HRs) and 95% confidence intervals (CIs) for all-cause and cause-specific mortality according to the quintiles of dietary vitamin B6.

	Quartile of Nutrient Intake				*P_trend_*
	Q1	Q2	Q3	Q4	
**Men**					
**All-cause mortality**					
Crude Model	Ref	0.76 (0.72–0.81)	0.65 (0.61–0.70)	0.57 (0.53–0.62)	<0.001
Multivariable Model	Ref	0.94 (0.88–1.00)	0.86 (0.80–0.93)	0.79 (0.71–0.86)	<0.001
**Cardiovascular mortality**					
Crude Model	Ref	0.71 (0.62–0.80)	0.55 (0.47–0.64)	0.51 (0.43–0.61)	<0.001
Multivariable Model	Ref	0.87 (0.76–0.98)	0.74 (0.63–0.86)	0.69 (0.56–0.85)	<0.001
**Cancer mortality**					
Crude Model	Ref	0.67 (0.59–0.76)	0.70 (0.61–0.80)	0.54 (0.45–0.64)	<0.001
Multivariable Model	Ref	0.85 (0.74–0.97)	0.95 (0.82–1.10)	0.73 (0.60–0.90)	0.013
**Women**					
**All-cause mortality**					
Crude Model	Ref	0.93 (0.87–0.99)	0.85 (0.79–0.91)	0.83 (0.76–0.91)	<0.001
Multivariable Model	Ref	0.93 (0.87–1.00)	0.90 (0.84–0.98)	0.88 (0.80–0.97)	0.002
**Cardiovascular mortality**					
Crude Model	Ref	0.89 (0.77–1.03)	0.71 (0.60–0.85)	0.58 (0.47–0.72)	<0.001
Multivariable Model	Ref	0.89 (0.77–1.03)	0.77 (0.64–0.92)	0.56 (0.44–0.73)	<0.001
**Cancer mortality**					
Crude Model	Ref	0.98 (0.84–1.13)	0.85 (0.71–1.00)	0.82 (0.68–1.00)	0.017
Multivariable Model	Ref	1.01 (0.87–1.17)	0.91 (0.76–1.08)	0.89 (0.72–1.10)	0.182

Crude HR did not adjust for anything. Multivariable HR adjusted for age, race/ethnicity, BMI, family income–poverty ratio, smoking status, drinking status, leisure-time physical activity, total energy intake, diabetes, hypertension, and cardiovascular disease.

**Table 4 nutrients-14-02253-t004:** Hazard Ratios (HRs) and 95% confidence intervals (CIs) for all-cause and cause-specific mortality according to the quintiles of dietary vitamin B12.

	Quartile of Nutrient Intake				*P_trend_*
	Q1	Q2	Q3	Q4	
**Men**					
**All-cause mortality**					
Crude Model	Ref	0.88 (0.82–0.93)	0.79 (0.73–0.84)	0.66 (0.61–0.72)	<0.001
Multivariable Model	Ref	1.01 (0.95–1.08)	1.01 (0.94–1.09)	1.01 (0.92–1.11)	0.719
**Cardiovascular mortality**					
Crude Model	Ref	0.78 (0.69–0.90)	0.68 (0.59–0.79)	0.56 (0.47–0.67)	<0.001
Multivariable Model	Ref	0.92 (0.80–1.05)	0.91 (0.77–1.07)	0.93 (0.76–1.13)	0.311
**Cancer mortality**					
Crude Model	Ref	0.84 (0.73–0.96)	0.85 (0.73–0.98)	0.65 (0.55–0.77)	<0.001
Multivariable Model	Ref	0.97 (0.84–1.11)	1.08 (0.93–1.27)	0.96 (0.79–1.16)	0.895
**Women**					
**All-cause mortality**					
Crude Model	Ref	0.88 (0.82–0.95)	0.83 (0.77–0.89)	0.77 (0.71–0.84)	<0.001
Multivariable Model	Ref	1.04 (0.97–1.11)	1.02 (0.94–1.11)	1.07 (0.97–1.18)	0.244
**Cardiovascular mortality**					
Crude Model	Ref	0.84 (0.72–0.97)	0.66 (0.56–0.79)	0.55 (0.45–0.68)	<0.001
Multivariable Model	Ref	1.11 (0.95–1.29)	0.98 (0.81–1.19)	1.05 (0.83–1.34)	0.848
**Cancer mortality**					
Crude Model	Ref	0.93 (0.80–1.08)	0.91 (0.77–1.07)	0.84 (0.69–1.01)	0.062
Multivariable Model	Ref	1.03 (0.88–1.20)	1.05 (0.88–1.26)	1.05 (0.84–1.31)	0.598

Crude HR did not adjust for anything. Multivariable HR adjusted for age, race/ethnicity, BMI, family income–poverty ratio, smoking status, drinking status, leisure-time physical activity, total energy intake, diabetes, hypertension, and cardiovascular disease.

## Data Availability

The data are publicly available online (https://wwwn.cdc.gov/nchs/nhanes/Default.aspx; accessed on 22 March 2022). The datasets used and analyzed during the current study are available from the corresponding author upon reasonable request.

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
