# Peer review of "Intakes of Folate, Vitamin B6, and Vitamin B12 in Relation to All-Cause and Cause-Specific Mortality: A National Population-Based Cohort"

_nutrients, 2022, doi:10.3390/nu14112253_

Round 1
Reviewer 1 Report
This paper describes associations between all-cause and cause-specific mortality and dietary folate, vitamin B6 and vitamin B12 in the general U.S. population (55,569 adults in NHANES III). They find an inverse association between folate and B6 with all-cause mortality, cardiovascular disease and cancer for men and with all-cause mortality and cardiovascular disease for women. They suggest increasing the intake of folate and vitamin B6 to lower mortality risk in U.S. adults.
Concerns:
It is stated that data were “prospectively evaluated” line 63. Survey data is retrospective.
A major limitation is lack of biochemical measurements of blood levels of the vitamins under study. More specifics should be provided in the Methods regarding how intake levels were determined from specific questions on the surveys. How is dietary intake of B6 and B12 being determined? Many foods contain both vitamins, but the associations here are different. The levels of folate, vitamin B6 and vitamin B12 in the 4 quartiles should be stated.
There are numerous types of cancer, and some are sex-specific, for example breast cancer in women. It would be beneficial to bin the cancer data by type and look at hazard ratios.
Data was used from NHANES surveys spanning 1988-1994 and 1999-2014. The NHANES sample population is representative of the U.S. population, but the food supply in the U.S. population drastically changed in the 1990s. A potentially confounding issue is the introduction of mandatory folic acid supplementation in the U.S. in 1996. Is folate intake differentiated from folic acid intake in this study? It would benefit the analyses to bin the data before/after the introduction of mandatory folic acid supplementation and look for differences.
Reviewer 2 Report
Intakes of folate, vitamin B6, and vitamin B12 in relation to all-cause and cause-specific mortality: A National Population-based Cohort is an interesting, well written paper, but I have some serious issues with background and methodology. The research is interesting, since the potential for vitamins, and more specifically B vitamins, in health promotion and disease prevention es important. Therefore, papers concerning the relation between vitamin intakes and mortality and health outcomes are warranted.
Authors have used data publicly available from the NHANES III and NHANES 1999-2014 National Survey. Nonetheless authors do not cite nor acknowledge previous researchers that provided data and made the present study available. There is a lot of work behind the NHANES survey, and it should be properly acknowledged and cited. Reference number 10 is incomplete and insufficient. Check the CDC websites for NHANES for suggested citation styles. Please provided some sentences highlighting the work done in NHANES and acknowledge previous work.
Introduction and references must be improved and updated. I do not believe there are only three relevant studies on the relation between folate and mortality, as stated in lines 48-27. Recheck the bibliographic review and search for recent papers and meta-analysis. For example, check the following:
Bo Y, Zhu Y, Tao Y, Li X, Zhai D, Bu Y, Wan Z, Wang L, Wang Y, Yu Z. Association Between Folate and Health Outcomes: An Umbrella Review of Meta-Analyses. Front Public Health. 2020 Dec 15;8:550753. doi: 10.3389/fpubh.2020.550753. PMID: 33384976; PMCID: PMC7770110.
It provides hundreds of studies on folate and mortality. Check for similar papers on vitamin B12 and Vitamin B6 in relation with mortality. Usually, information is scarcer in these two vitamins, but there are more published studies for sure. More references on previously published studies should be included.
Reference list includes only 3 out of 22 studies that have been published in 2020 or later, which is not adjusted to the relevance of the topic. More updated references, due to a more in-depth introduction will improve the paper. Discussion can also be enriched with updtated references.
Methodology should also be more extensively explained. For example, if data were de-identified, ¿how did the authors assess mortality by linking to the National Death Index? Please, explain.
Material and methods should be thoroughly revised to clearly state what part has been previously done by other researchers, providing clear references, and what part has been actually done by the authors. Eg. Lines 83 to 91: it is unclear whether authors have analysed the dietary records or these are already provided. Authors should avoid statements such as “information ….were collected through standardized questionnaires” “BMI was calculated as…”, “written informed consent was collected…” if not responsible for that part. Data selection criteria should be explained.
Minor revisions:
- Throughout the whole manuscript, please check for:
- Place final point or commas after references, not before
- Minor speeling mistakes and errors in English style
- Blank spaces
- Use commas for thousands, within text and tables
- Leave a blank space between HRs and 95%CI for a friendlier reading
- Tables: make them friendlier to read by separating data in different lines. Eg. HR in one line and 95%CI beneath
- Line 23: delete repeated “by”
- Line 30: delete “a” before cancer
- Line 44 and 45: substitute “are key contributors to” by “is key contributor”
- Line 58: check for grammar and style corrections.
- Line 73: substitute “participants” with “participant”
- Line 77: figure 1 is missing
- Line 103: replace “in formation” by “information”
- Table 1: add footnote explaining how results are expressed
- Lines 188-191: check the sentence. I do not understand whether there are differences or not.
- Line 192: delete the “a” before stronger
- Line 193: delete “the” before younger
- Line 205 and 2016: replace “fund” by “found”
- Lines 203-205: check the statement after updating bibliographic references
- Line 212: delete extra “has”
- Line 221-222: check grammar and style
- Line 251: replace “serous” by “series”
- Line 252: avoid repetition
- Lines 251-258: check verbal tenses
- Lines 259-266: the limitation applies to the NHANNES survey, not to the present study as such. For the present study, the limitation could be expressed as using pre-existing data that was obtained for a different purpose for a further study
- Line 295: explain what CENAN stands for
- Line 297-298: no statement on informed consent is necessary if using de-identified data. If compulsory to add, then add “by previous researchers” or similar.
Round 2
Reviewer 1 Report
The manuscript has been improved. A few issues remain.
Regarding original comment 2: how levels were determined has been added to the text, but please provide those estimated level ranges for each quartile based on the diet intake.
Regarding original comment 4: Table S9 should be referenced in the text with an explanation of the analysis and why it is important. Was the data in Table S9 the 1988-1994 or the 1999-2014 data, i.e. the yes or the no data as to folic acid supplementation of the food supply? Can both sets of data be shown? What is the N?
Reviewer 2 Report
Manusript has been improved. Thnk you. It only needs:
Minor English style and grammar corrections
Line 103. State you asked for and had permission to use the the restricted-use Linked Mortality File.